# Equivariant Graph Hierarchy-Based Neural Networks

**Jiaqi Han[1]**[*] **Wenbing Huang[2,3]**[†] **Tingyang Xu[4], Yu Rong[4]**[†]

[1] Department of Computer Science and Technology, Tsinghua University
[2] Gaoling School of Artificial Intelligence, Renmin University of China
[3] Beijing Key Laboratory of Big Data Management and Analysis Methods, Beijing, China
[4] Tencent AI Lab
alexhan99max@gmail.com, hwenbing@126.com
xuty007@gmail.com, yu.rong@hotmail.com

## Abstract

Equivariant Graph neural Networks (EGNs) are powerful in characterizing the dynamics of multi-body physical systems. Existing EGNs conduct *flat* message passing, which, yet, is unable to capture the spatial/dynamical hierarchy for complex systems particularly, limiting substructure discovery and global information fusion. In this paper, we propose Equivariant Hierarchy-based Graph Networks (EGHNs) which consist of the three key components: generalized Equivariant Matrix Message Passing (EMMP) , E-Pool and E-UnPool. In particular, EMMP is able to improve the expressivity of conventional equivariant message passing, E-Pool assigns the quantities of the low-level nodes into high-level clusters, while E-UnPool leverages the high-level information to update the dynamics of the low-level nodes. As their names imply, both E-Pool and E-UnPool are guaranteed to be E($n$)-equivariant to meet the physical symmetry. Considerable experimental evaluations verify the effectiveness of our EGHN on several applications including multi-object dynamics simulation, motion capture, and protein dynamics modeling.

## 1  Introduction

Understanding the multi-body physical systems is vital to numerous scientific problems, from microscopically how a protein with thousands of atoms acts and folds in the human body to macroscopically how celestial bodies influence each other's movement. While this is exactly an important form of expert intelligence, researchers have paid attention to teaching a machine to discover the physical rules from the observational systems through end-to-end trainable neural networks. Specifically, it is natural to use Graph Neural Networks (GNNs), which is able to model the relations between different bodies into a graph and the inter-body interaction as the message passing thereon [1, 16, 25, 26, 21].

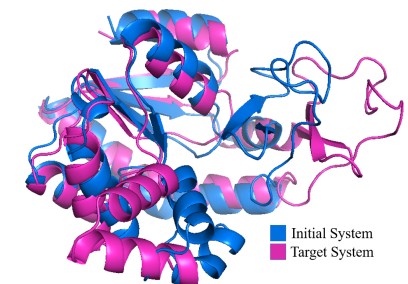

Figure 1: The folding dynamics of proteins in the cartoon format.

More recently, Equivariant GNNs (EGNs) [29, 8, 7, 27, 12] have become a crucial kind of tool for representing multi-body systems. One desirable property is that their outputs are equivariant with

---

[*]This work is done when Jiaqi Han works as an intern in Tencent AI Lab.
[†]Corresponding authors: Wenbing Huang and Yu Rong.

36th Conference on Neural Information Processing Systems (NeurIPS 2022).

respect to any translation/orientation/reflection of the inputs. With this inductive bias encapsulated, EGN permits the symmetry that the physical rules keep unchanged regardless of the reference coordinate system, enabling more enhanced generalization ability. Nevertheless, current EGNs only conduct *flat* message passing in the sense that each layer of message passing in EGN is formulated in the same graph space, where the spatial and dynamical information can only be propagated node-wisely and locally. By this design, it is difficult to discover the hierarchy of the patterns within complex systems.

*Hierarchy* is common in various domains. Imagine a complex mechanical system, where the particles are distributed on different rigid objects. In this case, for the particles on the same object, their states can be explained as the relative states to the object (probably the center) plus the dynamics of the object itself. We can easily track the behavior of the system if these "implicit" objects are detected automatically by the model we use. Another example, as illustrated in Figure 1, is the dynamics of a protein. Most proteins fold and change in the form of regularly repeating local structures, such as $\alpha$-helix, $\beta$-sheet and turns. By applying a hierarchical network, we are more capable of not only characterizing the conformation of a protein, but also facilitating the propagation between thousands of atoms in a protein by a more efficient means. There are earlier works proposed for hierarchical graph modeling [13, 5, 33, 3, 18], but these studies focus mainly on generic graph classification, and more importantly, they are not equivariant.

In this paper, we propose Equivariant Graph Hierarchy-based Network (EGHN), an end-to-end trainable model to discover local substructures of the input systems, while still maintaining the Euclidean equivariance. In a nutshell, EGHN is composed of an encoder and a decoder. The encoder processes the input system from fine-scale to coarse-scale, where an Equivariant-Pooling (E-Pool) layer is developed to group the low-level particles into each of a certain number of clusters that are considered as the particles of the next layer. By contrast, the decoder recovers the information from the coarse-scale system to the fine-scale one, by using the proposed Equivariant-Up-Pooling (E-UnPool) layer. Both E-Pool and E-UnPool are equivariant with regard to Euclidean transformations via our specific design. EGHN is built upon a generalized equivariant layer, which passes directional matrices over edges other than passing vectors in EGNN [27].

To verify the effectiveness of EGHN, we have simulated a new task extended from the N-body system [16], dubbed $M$-complex system, where each of the $M$ complexes is a rigid object comprised of a set of particles, and the dynamics of all complexes are driven by the electromagnetic force between particles. In addition to M-complex, we also carry out evaluations on two real applications: human motion caption [4] and the Molecular Dynamics (MD) of proteins [28]. For all tasks, our EGHN outperforms state-of-the-art EGN methods, indicating the efficacy and necessity of the proposed hierarchical modeling idea.[3]

## 2   Related Work

**GNNs for modeling physical interaction.** Graph Neural Networks (GNNs) have been widely investigated for modeling physical systems with multiple interacting objects. As pioneer attempts, Interaction Networks [1], NRI [16], and HRN [20] have been introduced to reason about the physical interactions. With the development of neural networks enforced by physical priors, many works resort to injecting physical knowledge into the design of GNNs. As an example, inspired by HNN [11], HOGN [25] models the evolution of interacting systems by Hamiltonian equations to obtain energy conservation. Another interesting feature of physical systems lies in Euclidean equivariance, *i.e.*, translation, rotation, and reflection. Several works first approach translation equivariance [30, 26, 21, 31]. Yet, dealing with rotation equivariance is non-trivial. TFN [29] and SE(3)-Transformer [8] leverages the irreducible representation of the SO(3) group, while LieConv [7] and LieTransformer [15] extend the realization of equivariance to Lie group. Apart from these works that resort to group representation theory, a succinct equivariant message passing scheme on E($n$) group is depicted in EGNN [27]. GMN [14] further involves equivariant forward kinematics modeling particularly for constrained systems. [2] generalizes EGNN to involve covariant information with steerable vectors. [22] leverages frame averaging for general equivariance. [19] mainly studies sign and basis invariance. Despite the rich literature, these models either violate the equivariance,

---

[3]Code is available at `https://github.com/hanjq17/EGHN`.

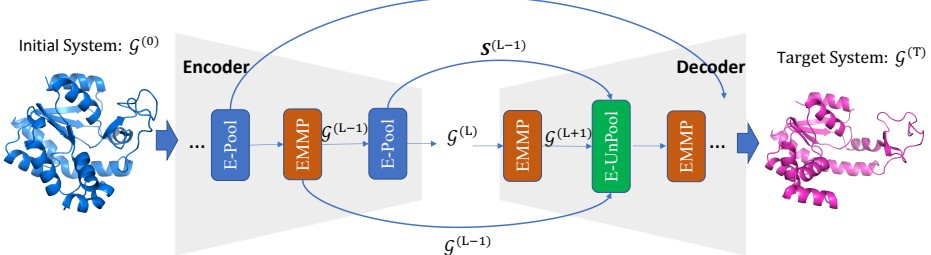

Figure 2: Illustration of the proposed EGHN. It consists of an encoder and a decoder, which are equipped with E-Pool and E-UnPool, respectively. E-UnPool takes as the input the previous output and the score matrix $\boldsymbol{S}$ from E-Pool and output the low-level system $\mathcal{G}$.

or inspect the system at a single granularity, both of which are vital aspects when tackling highly complicated systems like proteins.

**Hierarchical GNNs.** There are also works that explore the representation learning of GNNs in hierarchies. Several GNNs [13, 5, 32] adopt graph coarsening algorithms to view the graph in multiple granularities. [9] leverages a U-net architecture with top-$k$ pooling. Another line of work injects learnable pooling modules into the model. A differentiable pooling scheme DiffPool [33] has been introduced to learn a permutation-invariant pooling in an end-to-end manner. [3] replaces the aggregation in DiffPool by node dropping for saving the computational cost. [18] further incorporates self-attention mechanism into the pooling network. [6] leverages junction tree to model molecular graph in multiple hierarchies. Nevertheless, these techniques, although permutation equivariant, lack the guarantee of geometric equivariance, limiting their generalization on real-world 3D physical data.

## 3 The Proposed EGHN

This section first introduces the notations of our task, and then presents the design of the EMMP layer, which is the basic function in EGHN. Upon EMMP, the details of how the proposed E-Pool and E-UnPool work are provided. Finally, the instantiation of the entire architecture is described.

**Notations.** Each input multi-body system is modeled as a graph $\mathcal{G}$ consisting of $N$ particles (nodes) $\mathcal{V}$ and the interactions (edges) $\mathcal{E}$ among them. For each node $i$, it is assigned with a feature tuple $(\boldsymbol{Z}_i^{(0)}, \boldsymbol{h}_i^{(0)})$, where the directional matrix $\boldsymbol{Z}_i^{(0)} \in \mathbb{R}^{n \times m}$ is composed of $m$ $n$-dimension vectors, such as the concatenation of position $\boldsymbol{x}_i \in \mathbb{R}^3$ and velocity $\boldsymbol{v}_i \in \mathbb{R}^3$, leading to $\boldsymbol{Z}_i^{(0)} = [\boldsymbol{x}_i, \boldsymbol{v}_i] \in \mathbb{R}^{3 \times 2}$; $\boldsymbol{h}_i \in \mathbb{R}^c$ is the non-directional feature, such as the category of the atom in molecules. The edges are represented by an adjacency matrix $\boldsymbol{A} \in \mathbb{R}^{N \times N}$, which can either be constructed according to the geometric distance or physical connectivity. We henceforth abbreviate the entire information of a system, *i.e.*, $(\{\boldsymbol{Z}_i^{(0)}, \boldsymbol{h}_i^{(0)}\}_{i=1}^N, \boldsymbol{A})$ as the notation $\mathcal{G}^{\text{in}}$ if necessary.

We are mainly interested in investigating the dynamics of the input system $\mathcal{G}^{\text{in}}$. To be formal, given the initial state $(\boldsymbol{Z}_i^{(0)}, \boldsymbol{h}_i^{(0)})$ of each particle, our task is to find out a function $\phi$ to predict its future state $\boldsymbol{Z}_i^{(T)}$ given the interactions between particles. As explored before [29, 8, 7, 27], $\phi$ is implemented as a GNN to encode the inter-particle relation. In addition, it should be equivariant to any translation/reflection/rotation of the input states, so as to obey the physics symmetry about the coordinates. It means, $\forall g \in \mathrm{E}(n)$ that defines the Euclidean group [27],

$$\phi(\{g \cdot \boldsymbol{Z}_i^{(0)}\}_{i=1}^N, \cdots) = g \cdot \phi(\{\boldsymbol{Z}_i^{(0)}\}_{i=1}^N, \cdots), \tag{1}$$

where $g \cdot \boldsymbol{Z}_i^{(0)}$ conducts the orthogonal transformation as $\boldsymbol{R}\boldsymbol{Z}_i^{(0)}$ for both the position and velocity vectors and is additionally implemented as the translation $\boldsymbol{x}_i + \boldsymbol{b}$ for the position vector; the ellipsis denotes the input variables uninfluenced by $g$, including $\boldsymbol{h}_i^{(0)}$ and $\boldsymbol{A}$.

As discussed in Introduction, existing equivariant models [29, 8, 7, 27] are unable to mine the hierarchy within the dynamics of the input system by flat message passing. To address this pitfall, EGHN is formulated in the encoder-decoder form:

$$\mathcal{G}^{\text{high}} = \text{Encode}(\mathcal{G}^{\text{in}}), \mathcal{G}^{\text{out}} = \text{Decode}(\mathcal{G}^{\text{high}}, \mathcal{G}^{\text{in}}). \tag{2}$$

Here, as illustrated in Figure 2, the encoder aims at clustering the particles of $\mathcal{G}^{\text{in}}$ with similar dynamics into a group that is treated as the particle in the high-level graph $\mathcal{G}^{\text{high}}$ (the number of the nodes in $\mathcal{G}^{\text{high}}$ is smaller than $\mathcal{G}^{\text{in}}$). We have developed a novel component, E-Pool to fulfill this goal. As for the decoder, it recovers the information of all particles in the original graph space under the guidance of the high-level system $\mathcal{G}^{\text{high}}$, which is accomplished by the proposed E-UnPool. It is worth mentioning that both E-Pool and E-UnPool, as their names imply, are equivariant, and they are mainly built upon an expressive and generalized equivariant message passing layer, EMMP. To facilitate the understanding of our model, we first introduce the details of this layer in what follows.

## 3.1  Equivariant Matrix Message Passing

Given input features $\{(\boldsymbol{Z}_i, \boldsymbol{h}_i)\}_{i=1}^N$, EMMP performs information aggregation on the same graph to obtain the new features $\{(\boldsymbol{Z}'_i, \boldsymbol{h}'_i)\}_{i=1}^N$. The dimension of the output features could be different from the input, unless the row dimension of $\boldsymbol{Z}'_i$ should keep the same as $\boldsymbol{Z}_i$ (i.e. equal to $n$). In detail, one EMMP layer is updated by Eq. 3-6, where MLP($\cdot$) is a Multi-Layer Perceptron, $\mathcal{N}(i)$ collects the neighbors of $i$, and $\hat{\boldsymbol{Z}}_{ij} \in \mathbb{R}^{n \times 2m} = (\boldsymbol{Z}_i - \bar{\boldsymbol{Z}}, \boldsymbol{Z}_j - \bar{\boldsymbol{Z}})$ is a concatenation of the translated matrices on the edge $ij$. $\bar{\boldsymbol{Z}}$ is the mean of all nodes for the position vectors and zero for other vectors. With the subtraction of $\bar{\boldsymbol{Z}}$, $\hat{\boldsymbol{Z}}_{ij}$ is

$$\boldsymbol{H}_{ij}, \boldsymbol{h}_{ij} = \text{MLP}\left(\hat{\boldsymbol{Z}}_{ij}^\top \hat{\boldsymbol{Z}}_{ij}, \boldsymbol{h}_i, \boldsymbol{h}_j\right), \tag{3}$$

$$\boldsymbol{M}_{ij} = \hat{\boldsymbol{Z}}_{ij} \boldsymbol{H}_{ij}, \tag{4}$$

$$\boldsymbol{h}'_i = \text{MLP}\left(\boldsymbol{h}_i, \sum_{j \in \mathcal{N}(i)} \boldsymbol{h}_{ij}\right), \tag{5}$$

$$\boldsymbol{Z}'_i = \boldsymbol{Z}_i + \sum_{j \in \mathcal{N}(i)} \boldsymbol{M}_{ij}, \tag{6}$$

ensured to be translation invariant, and then $\boldsymbol{Z}'_i$ is translation equivariant after the addition of $\boldsymbol{Z}_i$ in Eq. 6. Specifically, the MLP in Eq. 3 takes as input the concatenation of the E($n$)-invariant $\hat{\boldsymbol{Z}}_{ij}^\top \hat{\boldsymbol{Z}}_{ij}, \boldsymbol{h}_i$, and $\boldsymbol{h}_j$, mapping from $\mathbb{R}^{2m \times 2m + 2c}$ to $\mathbb{R}^{2m \times m + c}$, and the output is split into $\boldsymbol{H}_{ij} \in \mathbb{R}^{2m \times m}$ and $\boldsymbol{h}_{ij} \in \mathbb{R}^c$. The formal proof for the E($n$)-equivariance of EMMP is deferred to Appendix.

Distinct from EGNN [27], the messages to pass in EMMP are directional matrices other than vectors. Although GMN [14] has also explored the matrix form, it is just a specific case of our EMMP by simplifying $\hat{\boldsymbol{Z}}_{ij} = \boldsymbol{Z}_i - \boldsymbol{Z}_j$. Indeed, we have the following theorem for the comparison of expressivity between EMMP, EGNN, and GMN, with the proof in Appendix.

**Theorem 1.** *EMMP can reduce to EGNN and GMN by specific choices of MLP in Eq. 3.*

Besides, since taking the inner product might induce a larger variance in the scale of input, in our implementation we also enforce a normalization $\hat{\boldsymbol{Z}}_{ij}^\top \hat{\boldsymbol{Z}}_{ij} / \|\hat{\boldsymbol{Z}}_{ij}^\top \hat{\boldsymbol{Z}}_{ij}\|_F$ before feeding the invariant $\hat{\boldsymbol{Z}}_{ij}^\top \hat{\boldsymbol{Z}}_{ij}$ into the MLP in Eq. 3, following the suggestion by GMN for better numerical stability.

## 3.2  Equivariant Pooling

Inspired by DiffPool, we propose E-Pool, an equivariant pooling module. Formally, E-Pool coarsens the low-level system $\mathcal{G}^{\text{low}} = (\{(\boldsymbol{Z}_i^{\text{low}}, \boldsymbol{h}_i^{\text{low}})\}_{i=1}^N, \boldsymbol{A}^{\text{low}})$ into an abstract and high-level system $\mathcal{G}^{\text{high}} = (\{(\boldsymbol{Z}_i^{\text{high}}, \boldsymbol{h}_i^{\text{high}})\}_{i=1}^K, \boldsymbol{A}^{\text{high}})$ with fewer particles, $K < N$. For this purpose, we first perform EMMP (Eq. 3-6) over the input system $\mathcal{G}$ to capture the local topology of each node. Then we apply the updated features of each node to predict which cluster it belongs to. This can be realized by a Softmax layer to output a soft score for each of the $K$ clusters. The cluster is deemed as a node of the high-level system, and its features are computed as a weighted combination of the low-level nodes with the scores it just derives. In summary, we proceed:

$$\{\boldsymbol{Z}'_i, \boldsymbol{h}'_i\}_i^N = \text{EMMP}(\{\boldsymbol{Z}_i^{\text{low}}, \boldsymbol{h}_i^{\text{low}}\}_i^N, \boldsymbol{A}^{\text{low}}), \tag{7}$$

$$\boldsymbol{s}_i = \text{Softmax}(\text{MLP}(\boldsymbol{h}'_i)), \tag{8}$$

$$\boldsymbol{Z}_j^{\text{high}} = \frac{1}{\sum_{i=1}^N s_{ij}} \sum_{i=1}^N s_{ij} \boldsymbol{Z}'_i, \tag{9}$$

$$\boldsymbol{h}_j^{\text{high}} = \frac{1}{\sum_{j=1}^N s_{ij}} \sum_{i=1}^N s_{ij} \boldsymbol{h}_i^{\text{low}}, \tag{10}$$

$$\boldsymbol{A}^{\text{high}} = \boldsymbol{S}^\top \boldsymbol{A}^{\text{low}} \boldsymbol{S}, \tag{11}$$

where Eq. 8 maps the invariant feature $\boldsymbol{h}'_i$ into the score $\boldsymbol{s}_i \in \mathbb{R}^K$ of cluster assignment with Softmax performed long the feature dimension, and the score matrix is given by $\boldsymbol{S} = [s_{ij}]_{N \times K}$ with $\boldsymbol{s}_i$ being its $i$-th row. By this design, it is tractable to verify that E-Pool is guaranteed to be E($n$) equivariant (also permutation equivariant). Specifically, the division by the row-wise sum $\sum_{i=1}^N s_{ij}$ in Eq. 9

is essential, as it permits the translation equivariance, that is, $\frac{1}{\sum_{i=1}^{N} s_{ij}} \sum_{i=1}^{N} s_{ij}(\boldsymbol{Z}_i' + \boldsymbol{b}) =$ $\left( \frac{1}{\sum_{i=1}^{N} s_{ij}} \sum_{i=1}^{N} s_{ij}\boldsymbol{Z}_i' \right) + \boldsymbol{b}$. This particular property distinguishes our pooling from traditional non-equivariant graph pooling [33, 18]. Notice that the normalization in Eq. 10 is unnecessary since $\boldsymbol{h}_i$ is a non-directional vector, but it is still adopted in line with Eq. 9. In practice, it is difficult to attain desirable clusters by using the SoftMax layer solely; instead, the pooling results are enhanced if we regulate the training process with an extra reconstruction loss related to the score matrix, whose formulation will be given in § 3.4.

## 3.3 Equivariant UnPooling

E-UnPool maps the information of the high-level system $\mathcal{G}^{\text{high}}$ back to the original system space $\mathcal{G}^{\text{low}}$, leading to an output system $\mathcal{G}^{\text{out}}$. We project the features back to the space of the original low-level system by using the transposed scores derived in E-Pool. Then, the projected features along with the low-level features are integrated by an E($n$) equivariant function to give the final output. The procedure of E-UnPool is given by Eq. 12-15, where $\hat{\boldsymbol{Z}}_i = [\boldsymbol{Z}_i^{\text{low}} - \bar{\boldsymbol{Z}}^{\text{low}}; \boldsymbol{Z}_i^{\text{agg}} - \bar{\boldsymbol{Z}}^{\text{agg}}]$ is the column-wise concatenation of the mean-translated low-level matrix $\boldsymbol{Z}_i^{\text{low}}$ and the high-level matrix $\boldsymbol{Z}_i^{\text{agg}}$, analogous to Eq. 3. One interesting point is that Eq. 12 is naturally equivariant in terms of translations, even without the normalization term used in Eq. 9. This is because the

$$\boldsymbol{Z}_i^{\text{agg}} = \sum_{j=1}^{K} s_{ij} \boldsymbol{Z}_j^{\text{high}}, \tag{12}$$

$$\boldsymbol{h}_i^{\text{agg}} = \sum_{j=1}^{K} s_{ij} \boldsymbol{h}_j^{\text{high}}, \tag{13}$$

$$\boldsymbol{h}_i^{\text{out}} = \text{MLP}\left( \hat{\boldsymbol{Z}}_i^{\top} \hat{\boldsymbol{Z}}_i, \boldsymbol{h}_i^{\text{low}}, \boldsymbol{h}_i^{\text{agg}} \right), \tag{14}$$

$$\boldsymbol{Z}_i^{\text{out}} = \hat{\boldsymbol{Z}}_i \boldsymbol{h}_i^{\text{out}} + \boldsymbol{Z}_i^{\text{agg}}, \tag{15}$$

score matrix is summed to 1 for each row, indicating that $\sum_{j=1}^{K} s_{ij}(\boldsymbol{Z}_j^{\text{high}} + \boldsymbol{b}) = \sum_{j=1}^{K} s_{ij}\boldsymbol{Z}_j^{\text{high}} + \boldsymbol{b}$. We have the following theorem guaranteeing the equivariance of the designed components, with all proofs deferred to Appendix.

**Theorem 2.** *EMMP, E-Pool, and E-UnPool are all E($n$)-equivariant.*

## 3.4 Instantiation of the Architecture

The overall architecture constitutes an equivariant U-Net [24] with skip-connections. We design the overall architecture as a sequence of EMMP, E-Pool, and E-UnPool in an encoder-decoder fashion, as depicted in Figure 2. The encoder is equipped with a certain number of E-Pools and EMMPs, while the decoder is realized with E-UnPools and EMMPs. For each E-UnPool in the decoder, as already defined in § 3.3, it is fed with the output of the previous layer, the score matrix $\boldsymbol{S}$ from E-Pool, and the low-level system $\mathcal{G}$ from EMMP in the corresponding layers of the encoder. Here, the so-called corresponding layers in E-Pool and E-UnPool are referred to the ones arranged in an inverse order; for example, in Figure 2, the final E-Pool corresponds to the first E-UnPool. With such design, it is straightforward, by the conclusion of Theorem 1, that the resulting EGHN still satisfies E($n$)-equivariance.

There is always one EMMP layer prior to each E-Pool or E-UnPool. This external EMMP plays a different role from the internal EMMP used in E-Pool (Eq. 7). One crucial difference is that they leverage different adjacency matrices. As we have introduced before, the adjacency matrix $\boldsymbol{A}$ can either be specified by geometric distance, *i.e.*, distance-based, or physical connectivity, *i.e.* connectivity-based. **1.** The external EMMP exploits a *distance-based* $\boldsymbol{A}_{\text{global}}$ whose element is valued if the distance between two particles is less than a threshold; by such means, we are able to characterize the force interaction between any two particles even they are physically disconnected. In higher-layer external EMMP, its $\boldsymbol{A}_{\text{global}}$ is created as a re-scored form (akin to Eq. 11) of $\boldsymbol{A}_{\text{global}}$ in lower layer, where the score matrix is obtained by its front E-Pool. **2.** For the internal EMMP in E-Pool, it applies a *connectivity-based* $\boldsymbol{A}_{\text{local}}$ that exactly reflects the physical connection between particles, for example, it is valued 1 if there is a bond between two atoms. In this way, E-Pool pays more attention to locally-connected particles when conducting clustering. Another minor point is that the external EMMP is relaxed as EGNN for only modeling the radial interaction, whereas the internal EMMP uses the generalized form in § 3.1. As we will show in our experiments, such design yields more favorable results compared with using any one of $\boldsymbol{A}_{\text{global}}$ and $\boldsymbol{A}_{\text{local}}$ only.

The training objective of EGHN is given by:

$$\mathcal{L} = \sum_{i=1}^{N} \|\boldsymbol{Z}_i^{\text{out}} - \boldsymbol{Z}_i^{\text{gt}}\|_F^2 + \lambda \sum_{l=1}^{L} \|(\boldsymbol{S}^{(l)})^\top \boldsymbol{A}^{(l-1)} \boldsymbol{S}^{(l)} - \boldsymbol{I}\|_F^2, \tag{16}$$

where $\|\cdot\|_F$ computes the Frobenius norm, $L$ is the number of E-Pools in the encoder, and $\lambda$ is the trade-off weight. The first term is to minimize the mean-square-error between the output state $\boldsymbol{Z}_i^{\text{out}}$ and the ground truth $\boldsymbol{Z}_i^{\text{gt}}$. The second term is the connectivity loss that encourages more connects within the pooling nodes and less cuts among pooling clusters [34]. For training stability, we first perform row-wise normalization of $(\boldsymbol{S}^{(l)})^\top \boldsymbol{A}^{(l-1)} \boldsymbol{S}^{(l)}$ before substituting it into Eq. 16.

## 4 Experiments

We contrast the performance of the proposed EGHN against a variety of baselines including the equivariant and non-equivariant GNNs, on one simulation task: the $M$-complex system, and the two real-world applications: human motion capture and molecular dynamics on proteins. We also carry out a complete set of ablation studies to verify the optimal design of our model.

### 4.1 Simulation Dataset: M-complex System

**Data generation.** We extend the $N$-body simulation system from [16] and generate the M-complex simulation dataset, in order to introduce hierarchical structures in the data. Specifically, we initialize a system with $N$ charged particles $\{\boldsymbol{x}_i, \boldsymbol{v}_i, c_i\}_{i=1}^{N}$ distributed on $M$ disjoint complex objects $\{\mathcal{S}_j\}_{j=1}^{M}$, where $\boldsymbol{x}_i, \boldsymbol{v}_i, c_i$ are separately the position, velocity, and charge for each particle. Within each complex $\mathcal{S}_j$, the particles are connected by rigid sticks, yielding geometric objects like sicks, triangles, tetrahedrons, etc. The dynamics of all $M$ complexes are driven by the electromagnetic force between every pair of particles. The task here is to predict the final positions $\{\boldsymbol{x}_i^T\}_i^N$ of all particles when $T = 1500$ given their initial positions and velocities. Without knowing which complex each particle belongs to, we will also test if our EGHN can group the particles correctly just based on the distribution of the trajectories. We independently sample $J$ different systems, each of which has $M$ complexes with the number of particles sampled from a uniform distribution with mean $N/M$. A dataset consits $J$ systems with $M$ complexes, $N/M$ average size of complex is abbreviated as $(M, N/M, J)$. We adopt Mean Squared Error (MSE) as the evaluation metric for the experiments.

**Implementation details.** We assign the node feature as the norm of the velocity $\|\boldsymbol{v}_i\|_2$, and the edge attribute as $c_i c_j$ for the edge connecting node $i$ and $j$, following the setting in [27]. We also concatenate an indicator, which is set as 1 if a stick presents and 0 otherwise, to the edge feature, similar to [14]. We use a fully connected graph (without self-loops) as $\boldsymbol{A}_{\text{global}}$, since the interaction force spans across each pair of particles in the system. The adjacency matrix $\boldsymbol{A}$ reflects the connectivity of the particles formed by the complexes. We set the number of clusters the same as the number of complexes in the dataset. The comparison models include: Linear Prediction (Linear) [27], SE(3)-Transformer (SE(3)-Tr.) [8], Radial-Field (RF) [17], MPNN [10] and EGNN [27]. For all these models, we employ the codes and architectures implemented by [27]. Detailed hyper-parameter settings are in Appendix.

**Results.** Table 1 reports the overall performance of the comparison models on eight simulation datasets with different configurations. From Table 1, we have the following observations: **1.** Clearly, EGHN surpasses all other approaches in all cases, demonstrating the general superiority of its design. **2.** Increasing the number of complexes ($M$) or the number of particles ($N$) always increases the complexity of the input system, but this does not necessarily hinder the performance of EGHN. For example, in both the single-system and multiple-system cases, EGHN even performs better when the system is changed from $(5, 5)$ to $(5, 10)$ and $(10, 10)$. We conjecture that, with more particles/complexes, larger systems also provide more data samples to enhance the training of EGHN. **3.** When increasing the diversity of systems ($J$) by switching from the single-system mode to multi-system mode, the performance of EGHN only drops slightly, indicating its adaptability to various scenarios. **Visualization.** we visualize in Figure 3 the predictions of EGNN and our EGHN on the $(3, 3, 1)$ scenario. We find that EGHN predicts the movements of the rigid objects more accurately than EGNN, especially for the large objects. In the right sub-figure, we also display the pooling results of EGHN, outputted by the score matrix of the final E-Pool layer. It is observed that EGHN is

Table 1: Prediction error ($\times 10^{-2}$) on various types of simulated datasets. The "Multiple System" contains $J = 5$ different systems. For each column, $(M, N/M)$ indicates that each system contains $M$ complexes of average size $N/M$. Results averaged across 3 runs. "OOM" denotes out of memory.

| | Single System | | | | Multiple Systems | | | |
| | (3, 3) | (5, 5) | (5, 10) | (10, 10) | (3, 3) | (5, 5) | (5, 10) | (10, 10) |
|---|---|---|---|---|---|---|---|---|
| Linear | $35.15_{\pm0.01}$ | $35.22_{\pm0.00}$ | $30.14_{\pm0.00}$ | $31.44_{\pm0.01}$ | $35.91_{\pm0.01}$ | $35.29_{\pm0.01}$ | $30.88_{\pm0.01}$ | $32.49_{\pm0.01}$ |
| TFN [29] | $25.11_{\pm0.15}$ | $29.35_{\pm0.17}$ | $26.01_{\pm0.22}$ | OOM | $27.33_{\pm0.21}$ | $29.01_{\pm0.13}$ | $25.57_{\pm0.14}$ | OOM |
| SE(3)-Tr. [8] | $27.12_{\pm0.26}$ | $28.87_{\pm0.09}$ | $24.48_{\pm0.35}$ | OOM | $28.14_{\pm0.16}$ | $28.66_{\pm0.10}$ | $25.00_{\pm0.28}$ | OOM |
| MPNN [10] | $16.00_{\pm0.11}$ | $17.55_{\pm0.19}$ | $16.15_{\pm0.08}$ | $15.91_{\pm0.15}$ | $16.76_{\pm0.13}$ | $17.58_{\pm0.11}$ | $16.55_{\pm0.21}$ | $16.05_{\pm0.16}$ |
| RF [17] | $14.20_{\pm0.09}$ | $18.37_{\pm0.12}$ | $17.08_{\pm0.03}$ | $18.57_{\pm0.30}$ | $15.17_{\pm0.10}$ | $18.55_{\pm0.12}$ | $17.24_{\pm0.11}$ | $19.34_{\pm0.25}$ |
| EGNN [27] | $12.69_{\pm0.19}$ | $15.37_{\pm0.13}$ | $15.12_{\pm0.11}$ | $14.64_{\pm0.27}$ | $13.33_{\pm0.12}$ | $15.48_{\pm0.16}$ | $15.29_{\pm0.12}$ | $15.02_{\pm0.18}$ |
| EGHN | $\mathbf{11.58}_{\pm0.01}$ | $\mathbf{14.42}_{\pm0.08}$ | $\mathbf{14.29}_{\pm0.40}$ | $\mathbf{13.09}_{\pm0.66}$ | $\mathbf{12.80}_{\pm0.56}$ | $\mathbf{14.85}_{\pm0.03}$ | $\mathbf{14.50}_{\pm0.08}$ | $\mathbf{13.11}_{\pm0.92}$ |

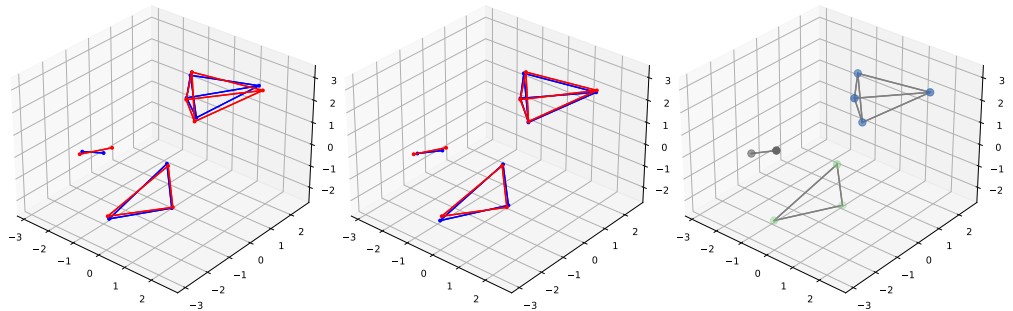

Figure 3: Visualization on M-complex systems. *Left*: the prediction of EGNN. *Middle*: the prediction of EGHN. *Right*: the pooling results of EGHN with each color indicating a cluster. In the left and middle figure, ground truth is in red, and prediction is in blue. Best viewed by colour printing.

able to detect the correct cluster for each particle. This is interesting and it can justify the worth of designing hierarchical architecture for multi-body system modeling.

## 4.2 Motion Capture

We further evaluate our model on CMU Motion Capture Databse [4]. We primarily focus on two activities, namely *walking* (Subject #35) [16] and *running* (Subject #9). With regard to walking, we leverage the random split adopted by [14], which includes 200 frame pairs for training, 600 for validation, and another 600 for testing. As for running, we follow a similar strategy and obtain a split with 200/240/240 frame pairs. The interval between each pair is 30 frames in both scenarios. In this task the joints are edges and their intersections are the nodes.

**Implementation details.** As discussed in [8], many real-world tasks, including our motion capture task here, break the Euclidean symmetry along the gravity axis ($z-$axis), and it is beneficial to make the equivariant models aware of where the top is. To this end, we augment the node feature by the coordinate of the $z-$axis, resulting in models that are height-aware while still equivariant in the horizontal directions. This operation is also applied to all baselines. Since the interaction of human body works along the joints, we propose to involve the edge in $A_{\text{global}}$ if it connects the nodes within two hops in $\mathcal{G}$. For the number of clusters $K$, we empirically find that $K = 5$ yields promising results for both walking and running.

Table 2: MSE ($\times 10^{-2}$) on the motion capture dataset averaged across 3 runs.

| | Subject #35 Walk | Subject #9 Run |
|---|---|---|
| MPNN [10] | 36.1 ±1.5 | 66.4 ±2.2 |
| RF [17] | 188.0 ±1.9 | 521.3 ±2.3 |
| TFN [29] | 32.0 ±1.8 | 56.6 ±1.7 |
| SE(3)-Tr. [8] | 31.5 ±2.1 | 61.2 ±2.3 |
| EGNN [27] | 28.7 ±1.6 | 50.9 ±0.9 |
| GMN [14] | 21.6 ±1.5 | 44.1 ±2.3 |
| EGHN | **8.5** ±2.2 | **25.9** ±0.3 |

**Results.** Table 2 summarizes the whole results of all models on two subjects. Here, we supplement an additional baseline GMN [14] for its promising performance on this task. Excitingly, EGHN outperforms all compared baselines by a large margin on both activities. Particularly, on Subject #35,

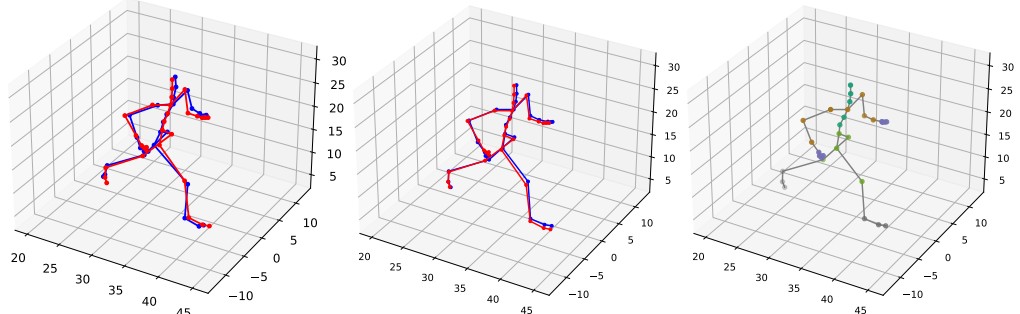

Figure 4: Visualization on Motion Capture. *Left*: the prediction of EGNN. *Middle*: the prediction of EGHN. *Right*: the pooling results of EGHN with each color indicating a cluster. In the left and middle figure, ground truth in red, and prediction in blue. Best viewed by zooming in.

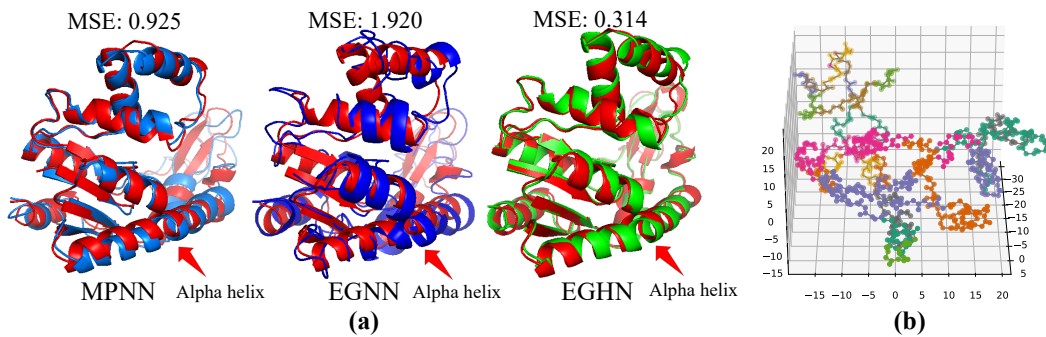

Figure 5: Visualization on the MDAnalysis dataset. (a) The predictions of MPNN, EGNN, and EGHN. Ground truth is in red. (b) The pooling assignment of EGHN.

the prediction error of EGHN is $8.5 \times 10^{-2}$, which is much lower than that of the best baseline, *i.e.*, GMN ($21.6 \times 10^{-2}$).

**Visualization.** To investigate why EGHN works, we depict the skeletons estimated by both EGNN and EGHN on Subject #9 in Figure 4. It shows that EGHN is able to capture more fine-grained details on certain parts (*e.g.* the junction between the legs and the body) than EGNN. When we additionally visualize the pooling outcome in the right sub-figure, we interestingly find that EGHN is capable of classifying the two right-left hands into the same cluster even they are spatially disconnected. A similar result is observed for the arms and feet. This is reasonable as EGHN checks not only if two particles are spatially close to each other but also if they share the similar dynamics.

### 4.3 Molecular Dynamics on Proteins

We adopt AdK equilibrium trajectory dataset [28] via MDAnalysis toolkit [23] to evaluate our hierarchical model. The AdK equilibrium trajectory dataset involves the MD trajectory of apo adenylate kinase simulated with explicit water and ions in NPT at 300 K and 1 bar. The atoms' positions of the protein are saved every 240 ps for a total of 1.004 $\mu$s as frames.

**Implementation details.** We split the dataset into train/validation/test sets along the timeline that contain 2481/827/878 frame pairs respectively. We choose $T = 15$ as the span between the input and prediction frames. We ignore the hydrogen atoms to focus on the prediction of large atoms. We further establish the global adjacency matrix as the neighboring atoms within a distance of 10Å. The atoms' velocities of the protein at each frame are computed by subtracting the positions to the previous frame's positions. We further leverage MDAnalysis to extract the protein backbone in order to reduce the data scale. Even so, TFN and SE(3)-Transformer still run out of memory, and thus we compare our model with the rest of baselines. Detailed hyper-parameters are in Appendix.

**Results.** The prediction MSE is depicted in Table 3. Our EGHN yields significantly lower error on protein MD compared with the baselines, achieving 1.843 MSE, while the sec-

Table 3: Prediction error (MSE) on protein MD.

| Linear | RF [17] | MPNN [10] | EGNN [27] | EGHN |
|--------|---------|-----------|-----------|------|
| 2.890  | 2.846   | 2.322     | 2.735     | **1.843** |

ond best model MPNN has an MSE of 2.322. However, MPNN is non-equivariant, and we find that its MSE will dramatically increase to 605.7 if we apply a random rotation of the protein during testing. Compared with EGNN, our EGHN exhibits its superiority thanks to the hierarchical modeling, particularly favorable on large and complex systems like proteins.

**Qualitative comparisons.** We visualize the protein structure of top-1 predictions generated by different models in cartoon format in Fig. 5 (a), with more visualization examples provided in Appendix. In Fig. 5 (a), the structure in red indicates the ground truth, while the other colors indicate the prediction. We can observe that EGHN tracks the folding and dynamics of the protein more precisely than the baselines. For example, in the the bottom region, EGHN gives a close-fitting result of the alpha helix structure while the predictions from MPNN and EGNN have an obvious shift compared with the ground truth. To validate the power of the E-Pool, we further visualize the pooling clusters in Fig. 5 (b). Interestingly, the pooling assignment exhibits certain clusters in some structures of the protein. It suggests that EGHN discovers local repetitive sub-structures of the protein; for instance, it detects the alpha helix structure in the middle of the protein.

## 4.4 Ablation Studies

We investigate the necessity of our proposed components on motion capture dataset in Table 4. We study the following questions:

**Q1.** *How will the performance of EGHN change, if we vary the number of clusters ($K$)?* We modify the number of clusters $K$ from 5 to 3 and 8, both of which yield worse performance. Specifically, we find that decreasing $K$ on "Run" results in a larger degradation of performance, possibly because the activity "Run" is with complicated kinematics and it will be more difficult to learn if the joints are shared across a too small number of clusters. We provide potential guidance on choosing $K$ in Appendix D.1. **Q2.** *How do our proposed two components EMMP and hi-*

Table 4: Ablation studies on the motion capture dataset. Numbers are MSE ($\times 10^{-2}$).

|  | Subject #35 Walk | Subject #9 Run |
|--|------------------|----------------|
| EGHN ($K = 5$) | **8.5** | **25.9** |
| EGHN ($K = 3$) | 10.1 | 41.4 |
| EGHN ($K = 8$) | 14.9 | 26.8 |
| w/o Equivariance | 19.7 | 40.9 |
| w/o Hierarchy | 21.9 | 42.1 |
| Replace by EGNN | 22.3 | 42.5 |
| w/o Connectivity loss | 10.5 | 28.8 |
| $A_{\text{global}}$ only | 17.4 | 31.5 |
| $A_{\text{local}}$ only | 16.8 | 33.5 |

*erarchical modeling contribute?* We replace all EMMP layers in our model by typical non-equivariant MPNN, and the performance drops from 8.5 to 19.7 on Walk, supporting that maintaining equivariance is vital. We further set $s_i = \mathbf{1}_i$ in all E-Pool and E-UnPool and observe that removing hierarchy is detrimental to accurate prediction. Moreover, by replacing all EMMPs with EGNNs, the performance also drops, which aligns with our analysis on the stronger expressivity of EMMP over EGNN. Complete studies are deferred to Appendix D.2 and D.3. **Q3.** *How does the connectivity loss (the second term in Eq. 16) help?* By dropping the connectivity loss, we observe a larger prediction error. This justifies the necessity of using the connectivity loss to focus more on intra-cluster connections against the inter-cluster edges. **Q4.** *How about using the same adjacency matrix for all EMMP instead of distinguishing them as $A_{global}$ in the external EMMPs and $A_{local}$ in internal EMMPs as discussed in § 3.4?* When we apply $A_{\text{global}}$ or $A_{\text{local}}$ for all EMMPs, the performance drops dramatically, implying that the external and internal EMMPs play different roles in EGHN, and should be equipped with different adjacency matrices to model the interactions of different scopes.

## 5 Discussion

**Limitation.** In the current form the number of clusters $K$ is fixed in EGHN as an empirical hyperparameter. Future works include extending E-Pool to dynamically adjust $K$ for systems with different scales for enhancing the flexibility of the hierarchical model.

**Conclusion.** In this paper, we propose Equivariant Graph Hierarchy-based Network (EGHN) to model and represent the dynamics of multi-body systems. EGHN leverages E-Pool to group the low-level nodes into clusters, and E-UnPool to restore the low-level information from the high-level systems with the aid of the corresponding E-Pool layer. The fundamental layer of EGHN lies in Equivariant Matrix Message Passing (EMMP) to characterize the topology and dynamics expressively. Experimental evaluations on M-complex systems, Motion-Capture, and protein MD, show that EGHN consistently outperforms other non-hierarchical EGNs as well as non-equivariant GNNs.

## Acknowledgments and Disclosure of Funding

We thank the anonymous reviewers for their helpful suggestions. This work was jointly supported by the following projects: the Scientific Innovation 2030 Major Project for New Generation of AI under Grant No. 2020AAA0107300, Ministry of Science and Technology of the People's Republic of China; the National Natural Science Foundation of China (No.62006137); Guoqiang Research Institute General Project, Tsinghua University (No. 2021GQG1012); Beijing Outstanding Young Scientist Program (No. BJJWZYJH012019100020098); Tencent AI Lab Rhino-Bird Visiting Scholars Program (VS2022TEG001).

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
