# Supplementary Materials for
# Equivariant Graph Hierarchy-Based Neural Networks

## Contents

## A Proofs

### A.1 Proof of Theorem 1

**Theorem 1.** *EMMP can reduce to EGNN and GMN by specific choices of MLP in Eq. 3.*

*Proof.* For simplicity, we denote $\boldsymbol{Z}_i - \bar{\boldsymbol{Z}}$ as $\bar{\boldsymbol{Z}}_i$, which infers $\bar{\boldsymbol{Z}}_i - \bar{\boldsymbol{Z}}_j = \boldsymbol{Z}_i - \boldsymbol{Z}_j$.

For EMMP, GMN [4], and EGNN [6], we rewrite their messages (Eq. 3-4) below.

$$\boldsymbol{M}_{ij}^{\mathrm{EMMP}} = \hat{\boldsymbol{Z}}_{ij}\mathrm{MLP}_1\left(\hat{\boldsymbol{Z}}_{ij}^\top \hat{\boldsymbol{Z}}_{ij}\right),$$

$$= \begin{bmatrix} \bar{\boldsymbol{Z}}_i & \bar{\boldsymbol{Z}}_j \end{bmatrix}\mathrm{MLP}_1\left(\begin{bmatrix} \bar{\boldsymbol{Z}}_i^\top \bar{\boldsymbol{Z}}_i & \bar{\boldsymbol{Z}}_i^\top \bar{\boldsymbol{Z}}_j \\ \bar{\boldsymbol{Z}}_j^\top \bar{\boldsymbol{Z}}_i & \bar{\boldsymbol{Z}}_j^\top \bar{\boldsymbol{Z}}_j \end{bmatrix}\right).$$

$$\boldsymbol{M}_{ij}^{\mathrm{GMN}} = (\boldsymbol{Z}_i - \boldsymbol{Z}_j)\mathrm{MLP}_2\left((\boldsymbol{Z}_i - \boldsymbol{Z}_j)^\top(\boldsymbol{Z}_i - \boldsymbol{Z}_j)\right).$$

$$\boldsymbol{M}_{ij}^{\mathrm{EGNN}} = (\boldsymbol{x}_i - \boldsymbol{x}_j)\mathrm{MLP}_3\left((\boldsymbol{x}_i - \boldsymbol{x}_j)^\top(\boldsymbol{x}_i - \boldsymbol{x}_j)\right).$$

**1.** We first prove that EMMP can reduce to GMN.

Let $\text{MLP}_1 = f_{\text{out}} \circ \text{MLP}_2 \circ f_{\text{in}}$, where $f_{\text{in}}(\begin{bmatrix} \boldsymbol{a}_{11} & \boldsymbol{a}_{12} \\ \boldsymbol{a}_{21} & \boldsymbol{a}_{22} \end{bmatrix}) = (\boldsymbol{a}_{11} - \boldsymbol{a}_{12}) - (\boldsymbol{a}_{21} - \boldsymbol{a}_{22})$, $f_{\text{out}}(\boldsymbol{a}) = \begin{bmatrix} \boldsymbol{a} \\ -\boldsymbol{a} \end{bmatrix}$, and "∘" is the function composition. By this relaxation, EMMP reduces to:

$$
\begin{aligned}
\boldsymbol{M}_{ij}^{\text{EMMP}} &= \begin{bmatrix} \bar{\boldsymbol{Z}}_i & \bar{\boldsymbol{Z}}_j \end{bmatrix} f_{\text{out}} \circ \text{MLP}_2 \circ f_{\text{in}} \left( \begin{bmatrix} \bar{\boldsymbol{Z}}_i^\top \bar{\boldsymbol{Z}}_i & \bar{\boldsymbol{Z}}_i^\top \bar{\boldsymbol{Z}}_j \\ \bar{\boldsymbol{Z}}_j^\top \bar{\boldsymbol{Z}}_i & \bar{\boldsymbol{Z}}_j^\top \bar{\boldsymbol{Z}}_j \end{bmatrix} \right), \\
&= \begin{bmatrix} \bar{\boldsymbol{Z}}_i & \bar{\boldsymbol{Z}}_j \end{bmatrix} f_{\text{out}} \circ \text{MLP}_2 \left( \bar{\boldsymbol{Z}}_i^\top (\bar{\boldsymbol{Z}}_i - \bar{\boldsymbol{Z}}_j) - \bar{\boldsymbol{Z}}_j^\top (\bar{\boldsymbol{Z}}_i - \bar{\boldsymbol{Z}}_j) \right), \\
&= \begin{bmatrix} \bar{\boldsymbol{Z}}_i & \bar{\boldsymbol{Z}}_j \end{bmatrix} f_{\text{out}} \left( \text{MLP}_2 \left( (\boldsymbol{Z}_i - \boldsymbol{Z}_j)^\top (\boldsymbol{Z}_i - \boldsymbol{Z}_j) \right) \right), \\
&= \begin{bmatrix} \bar{\boldsymbol{Z}}_i & \bar{\boldsymbol{Z}}_j \end{bmatrix} \begin{bmatrix} \text{MLP}_2 \left( (\boldsymbol{Z}_i - \boldsymbol{Z}_j)^\top (\boldsymbol{Z}_i - \boldsymbol{Z}_j) \right) \\ -\text{MLP}_2 \left( (\boldsymbol{Z}_i - \boldsymbol{Z}_j)^\top (\boldsymbol{Z}_i - \boldsymbol{Z}_j) \right) \end{bmatrix}, \\
&= (\boldsymbol{Z}_i - \boldsymbol{Z}_j) \text{MLP}_2 \left( (\boldsymbol{Z}_i - \boldsymbol{Z}_j)^\top (\boldsymbol{Z}_i - \boldsymbol{Z}_j) \right), \\
&= \boldsymbol{M}_{ij}^{\text{GMN}}.
\end{aligned}
$$

**2.** We then prove that GMN can reduce to EGNN using similar derivations as above.

Denote $\boldsymbol{Z}_i = [\boldsymbol{x}_i, \boldsymbol{v}_i]$, and we can similarly let $\text{MLP}_2 = f_{\text{out}} \circ \text{MLP}_3 \circ f_{\text{in}}$, where $f_{\text{in}}(\begin{bmatrix} \boldsymbol{a}_{11} & \boldsymbol{a}_{12} \\ \boldsymbol{a}_{21} & \boldsymbol{a}_{22} \end{bmatrix}) = \boldsymbol{a}_{11}$, and $f_{\text{out}}(\boldsymbol{a}) = \begin{bmatrix} \boldsymbol{a} \\ \boldsymbol{0} \end{bmatrix}$. Therefore, we have that

$$
\begin{aligned}
\boldsymbol{M}_{ij}^{\text{GMN}} &= \begin{bmatrix} \boldsymbol{x}_i - \boldsymbol{x}_j & \boldsymbol{v}_i - \boldsymbol{v}_j \end{bmatrix} f_{\text{out}} \circ \text{MLP}_3 \circ f_{\text{in}} \left( \begin{bmatrix} (\boldsymbol{x}_i - \boldsymbol{x}_j)^\top (\boldsymbol{x}_i - \boldsymbol{x}_j) & (\boldsymbol{x}_i - \boldsymbol{x}_j)^\top (\boldsymbol{v}_i - \boldsymbol{v}_j) \\ (\boldsymbol{v}_i - \boldsymbol{v}_j)^\top (\boldsymbol{x}_i - \boldsymbol{x}_j) & (\boldsymbol{v}_i - \boldsymbol{v}_j)^\top (\boldsymbol{v}_i - \boldsymbol{v}_j) \end{bmatrix} \right), \\
&= \begin{bmatrix} \boldsymbol{x}_i - \boldsymbol{x}_j & \boldsymbol{v}_i - \boldsymbol{v}_j \end{bmatrix} f_{\text{out}} \circ \text{MLP}_3 \left( (\boldsymbol{x}_i - \boldsymbol{x}_j)^\top (\boldsymbol{x}_i - \boldsymbol{x}_j) \right), \\
&= \begin{bmatrix} \boldsymbol{x}_i - \boldsymbol{x}_j & \boldsymbol{v}_i - \boldsymbol{v}_j \end{bmatrix} \begin{bmatrix} \text{MLP}_3 \left( (\boldsymbol{x}_i - \boldsymbol{x}_j)^\top (\boldsymbol{x}_i - \boldsymbol{x}_j) \right) \\ \boldsymbol{0} \end{bmatrix}, \\
&= (\boldsymbol{x}_i - \boldsymbol{x}_j) \text{MLP}_3 \left( (\boldsymbol{x}_i - \boldsymbol{x}_j)^\top (\boldsymbol{x}_i - \boldsymbol{x}_j) \right), \\
&= \boldsymbol{M}_{ij}^{\text{EGNN}},
\end{aligned}
$$

which concludes the proof. $\square$

This theorem basically implies that the expressivity of our EMMP is stronger than that of GMN or EGNN.

### A.2 Proof of Theorem 2

**Theorem 2.** *EMMP, E-Pool, and E-UnPool are all E(n)-equivariant.*

*Proof.* **1.** We first prove that EMMP is E($n$)-equivariant.

For any $g \in \text{E}(n)$, we have $g \cdot \boldsymbol{Z} = \boldsymbol{R}\boldsymbol{Z} + \boldsymbol{b}$ where $\boldsymbol{R} \in \mathbb{R}^{3 \times 3}$, $\boldsymbol{R}^\top \boldsymbol{R} = \boldsymbol{I}$ and $\boldsymbol{b} \in \mathbb{R}^3$. We use the superscript $*$ to denote the resulting output after applying the group action $g$ to the input. Initially, we have $\boldsymbol{Z}^* = \boldsymbol{R}\boldsymbol{Z} + \boldsymbol{b}$, and $\boldsymbol{h}_i^* = \boldsymbol{h}_i$. Similarly, $\bar{\boldsymbol{Z}}^* = \boldsymbol{R}\bar{\boldsymbol{Z}} + \boldsymbol{b}$. We proceed the proof step by step, following the definition of EMMP in Eq. 3-6:

$$
\begin{aligned}
\hat{\boldsymbol{Z}}_{ij}^* &= [\boldsymbol{Z}_i^* - \bar{\boldsymbol{Z}}^*, \boldsymbol{Z}_j^* - \bar{\boldsymbol{Z}}^*] = [\boldsymbol{R}\boldsymbol{Z}_i + \boldsymbol{b} - (\boldsymbol{R}\bar{\boldsymbol{Z}} + \boldsymbol{b}), \boldsymbol{R}\boldsymbol{Z}_j + \boldsymbol{b} - (\boldsymbol{R}\bar{\boldsymbol{Z}} + \boldsymbol{b})] = \boldsymbol{R}\hat{\boldsymbol{Z}}_{ij}, \\
\boldsymbol{H}_{ij}^* &= \text{MLP} \left( (\boldsymbol{R}\hat{\boldsymbol{Z}}_{ij})^\top \boldsymbol{R}\hat{\boldsymbol{Z}}_{ij}, \boldsymbol{h}_i, \boldsymbol{h}_j \right) = \text{MLP} \left( \hat{\boldsymbol{Z}}_{ij}^\top \hat{\boldsymbol{Z}}_{ij}, \boldsymbol{h}_i, \boldsymbol{h}_j \right) = \boldsymbol{H}_{ij}, \\
\boldsymbol{M}_{ij}^* &= \hat{\boldsymbol{Z}}_{ij}^* \boldsymbol{H}_{ij}^* = \boldsymbol{R}\hat{\boldsymbol{Z}}_{ij}\boldsymbol{H}_{ij} = \boldsymbol{R}\boldsymbol{M}_{ij}, \\
\boldsymbol{h}_i'^* &= \text{MLP} \left( \boldsymbol{h}_i, \sum_{j \in \mathcal{N}(i)} \boldsymbol{H}_{ij} \right) = \boldsymbol{h}_i', \\
\boldsymbol{Z}_i'^* &= \boldsymbol{R}\boldsymbol{Z} + \boldsymbol{b} + \sum_{j \in \mathcal{N}(i)} \boldsymbol{R}\boldsymbol{M}_{ij} = \boldsymbol{R}(\boldsymbol{Z} + \sum_{j \in \mathcal{N}(i)} \boldsymbol{M}_{ij}) + \boldsymbol{b} = \boldsymbol{R}\boldsymbol{Z}_i' + \boldsymbol{b},
\end{aligned}
$$

which verifies that EMMP is E($n$)-equivariant.

**2.** We then prove that E-Pool is E($n$)-equivariant.

$$\boldsymbol{Z}_j^{\text{high},*} = \frac{1}{\sum_{i=1}^{N} s_{ij}} \sum_{i=1}^{N} s_{ij}(\boldsymbol{R}\boldsymbol{Z}_i' + \boldsymbol{b}) = \boldsymbol{R}(\frac{1}{\sum_{i=1}^{N} s_{ij}} \sum_{i=1}^{N} s_{ij}\boldsymbol{Z}_i') + \boldsymbol{b} = \boldsymbol{R}\boldsymbol{Z}_j^{\text{high}} + \boldsymbol{b},$$

$$\boldsymbol{h}_j^{\text{high},*} = \frac{1}{\sum_{j=1}^{N} s_{ij}} \sum_{i=1}^{N} s_{ij}\boldsymbol{h}_i^{\text{low}} = \boldsymbol{h}_j^{\text{high}},$$

$$\boldsymbol{A}^{\text{high},*} = \boldsymbol{S}^{\top}\boldsymbol{A}^{\text{low}}\boldsymbol{S} = \boldsymbol{A}^{\text{high}},$$

which clearly shows that E-Pool is E($n$)-equivariant, while the high-level adjacency matrix $\boldsymbol{A}^{\text{high}}$ is E($n$)-invariant, which is crucial for maintaining the equivariance of the high-level EMMP.

**3.** Finally we prove that E-UnPool is E($n$)-equivariant.

$$\boldsymbol{Z}_i^{\text{agg},*} = \sum_{j=1}^{K} s_{ij}(\boldsymbol{R}\boldsymbol{Z}_j^{\text{high}} + \boldsymbol{b}) = \boldsymbol{R}(\sum_{j=1}^{K} s_{ij}\boldsymbol{Z}_j^{\text{high}}) + \boldsymbol{b} = \boldsymbol{R}\boldsymbol{Z}_i^{\text{agg}} + \boldsymbol{b},$$

$$\boldsymbol{h}_i^{\text{agg},*} = \boldsymbol{h}_i^{\text{agg}},$$

$$\boldsymbol{h}_i^{\text{out},*} = \text{MLP}\left((\boldsymbol{R}\hat{\boldsymbol{Z}}_i)^{\top}(\boldsymbol{R}\hat{\boldsymbol{Z}}_i), \boldsymbol{h}_i^{\text{low}}, \boldsymbol{h}_i^{\text{agg}}\right) = \boldsymbol{h}_i^{\text{out}},$$

$$\boldsymbol{Z}_i^{\text{out},*} = \boldsymbol{R}\hat{\boldsymbol{Z}}_i\boldsymbol{h}_i^{\text{out}} + \boldsymbol{R}\boldsymbol{Z}_i^{\text{agg}} + \boldsymbol{b} = \boldsymbol{R}\boldsymbol{Z}_i^{\text{out}} + \boldsymbol{b}.$$

$\square$

Indeed, with Theorem 2 we immediately have that any cascade of EMMP, E-Pool, and E-UnPool is also E($n$)-equivariant. This indicates that our resulting EGHN is E($n$)-equivariant.

## B  Implementation Details

**Baselines.** For the baselines, we leverage the codebases maintained by [4][1] and [6][2], which are released under MIT license. We tune the hyper-parameters around the suggested hyper-parameters as specified in [4] and [6] for the baselines. Specifically, for MPNN [3], RF [5] and EGNN [6], we tune the learning rate from {1e-4, 5e-4, 1e-3}, weight decay {1e-12, 1e-10, 1e-8, 1e-4}, batch size {50, 100, 200}, hidden dim {32, 64, 128} and the number of layers {2, 4, 6, 8}. For TFN [7] and SE(3)-Transformer [2], we set the degree to 2 due to memory limitation, and select the learning rate from {5e-4, 1e-3, 5e-3}, weight decay {1e-10, 1e-8}, batch size {25, 50, 100}, hidden dim {32, 64} and the number of layers {2, 4}. We report the best results searched within these ranges of hyper-parameters for the baselines. We use an early-stopping of 50 epochs for all methods. Note that the kinematics decomposition trick in GMN [4] requires a specific design to enforce hard constraints for any new system, which cannot be directly applied to our simulation dataset and protein MD. Besides, both TFN and SE(3)-Transformer run out of memory on protein MD, and we thus omit their results in Table 3.

**EGHN.** For our EGHN, on simulation dataset, we use batch size 50, and the number of clusters the same as the complexes in the dataset. On motion capture, we use batch size 12, and the number of clusters $K = 5$ on both datasets. On MD dataset, we use batch size 8, and the number of clusters $K = 15$. Table 5 depicts the rest of detailed hyper-parameter configurations. Notably, to control the computational budget of EGHN compared with the baselines, we set the maximum number of encoder/decoder layers as 4, while for the baselines we set the maximum number of layers as 8, ensuring fair comparison. All experiments are conducted on NVIDIA Tesla V100 GPU.

Besides, to gain more insights of our design of $\boldsymbol{A}_{\text{global}}$ and $\boldsymbol{A}_{\text{local}}$, we provide an illustration in Fig. 6. Our intuition is that the relation modeling in different hierarchy levels might contain different

---

[1] https://github.com/hanjq17/GMN
[2] https://github.com/vgsatorras/egnn

Table 5: Hyper-parameters of EGHN.

| Dataset | learning rate | $\lambda$ | weight decay | Encoder Layer | Decoder Layer |
|---------|---------------|-----------|--------------|---------------|---------------|
| (3, 3, 1) | 0.0005 | 4 | 1e-4 | 4 | 2 |
| (3, 3, 5) | 0.001 | 4 | 1e-4 | 4 | 2 |
| (5, 5, 1) | 0.0003 | 2 | 1e-6 | 4 | 2 |
| (5, 5, 5) | 0.001 | 0.1 | 1e-12 | 4 | 2 |
| (5, 10, 1) | 0.0001 | 4 | 1e-4 | 2 | 2 |
| (5, 10, 5) | 0.0005 | 4 | 1e-4 | 4 | 2 |
| (10, 10, 1) | 0.0005 | 2 | 1e-6 | 4 | 2 |
| (10, 10, 5) | 0.0003 | 1 | 1e-8 | 4 | 2 |
| Mocap Walk | 0.0004 | 1 | 1e-6 | 2 | 2 |
| Mocap Run | 0.0003 | 1 | 1e-6 | 4 | 1 |
| MD | 0.0005 | 0.1 | 1e-8 | 4 | 2 |

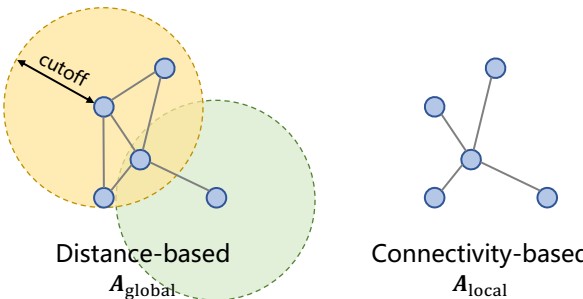

Figure 6: An illustration of $\boldsymbol{A}_{\mathrm{global}}$ and $\boldsymbol{A}_{\mathrm{local}}$.

semantics. For example, in the external EMMP, we use $\boldsymbol{A}_{\mathrm{global}}$ since we would like the model to capture and gather the interaction forces based on the distance between nodes (atoms). As for the internal EMMP, the topology of the graph, *i.e.*, the connectivity, plays an important role in determining the topological information (such as the bond connection in molecules and proteins) which is crucial for performing pooling and unpooling. Our connectivity loss, by sharing a similar idea, also enforces a stronger connectivity on the pooling assignment by encouraging connected nodes to be pooled into the same cluster and penalizing the others. By this design, EGHN is designed to be more flexible and the ablations also verify the efficacy of leveraging $\boldsymbol{A}_{\mathrm{global}}$ and $\boldsymbol{A}_{\mathrm{local}}$ in external and internal EMMP, respectively.

Furthermore, in order to keep a fair comparison between EGHN and the baselines, we augment the edge feature of the baselines by taking into account the information of $\boldsymbol{A}_{\mathrm{global}}$ and $\boldsymbol{A}_{\mathrm{local}}$. Specifically, for the set of edges we employ $\boldsymbol{A}_{\mathrm{global}}$, while extending a channel on the edge feature by an indicator function that takes the value 1 if this edge also belongs to $\boldsymbol{A}_{\mathrm{local}}$ and 0 otherwise. On all the three datasets, it is satisfied that $\boldsymbol{A}_{\mathrm{local}}$ is always a subset of $\boldsymbol{A}_{\mathrm{global}}$ by our choices. Therefore, through such augmentation, we exactly keep the same edge information between EGHN and baselines without any unfairness.

**More explanations on the connectivity loss.** Intuitively, the connectivity loss encourages pooling assignments with more edges within the pooled clusters and fewer in between. In particular, the loss reaches its minimum, *i.e.*, 0, if and only if node $i$ and $j$ belong to the same cluster for each edge $(i, j) \in \mathcal{E}$.

## C Learning Curve

We provide the learning curve of EGHN and EGNN on (3, 3, 1) of the $M$-complex dataset. It is illustrated that EGHN converges faster and the corresponding testing loss is lower as well, yielding better performance than EGNN.

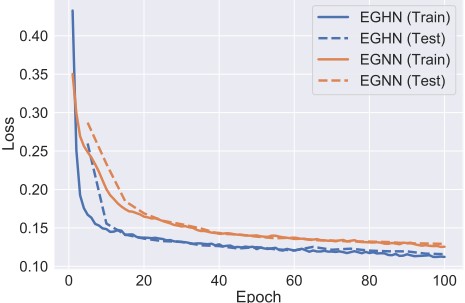

Figure 7: The learning curves of EGHN and EGNN on (3, 3, 1) of the $M$-complex dataset.

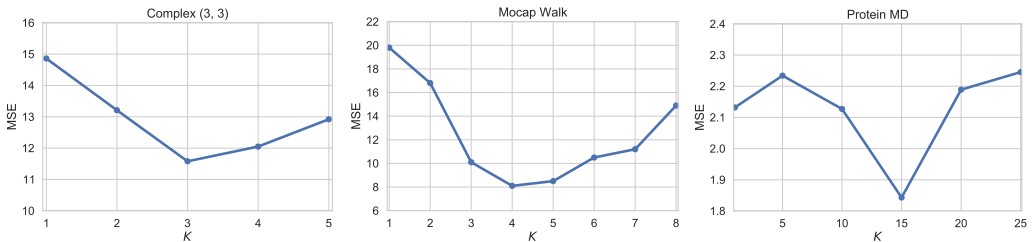

Figure 8: Prediction MSE *w.r.t.* the number of clusters $K$.

## D  More ablation studies

### D.1  The impact of the number of clusters $K$

We thoroughly investigate how the number of clusters influence the model performance on all datasets. For $M$-complex System, we sweep over 1 to 5 in the Complex (3, 3) single system. For Mocap dataset, we sweep over 1 to 8. For Protein MD, we vary $K$ from 1, 5, 10, 15, 20, 25. The results are depicted in Table 6, 7, and 8. We also provide the number of nodes of each system in these tables. A visualization can be found in Fig. 8.

Table 6: MSE ($\times 10^{-2}$) on Complex (3, 3) *w.r.t.* the number of clusters $K$.

| 9 nodes | 1 | 2 | 3 | 4 | 5 |
|---------|-------|-------|--------|-------|-------|
| MSE | 14.86 | 13.21 | **11.58** | 12.05 | 12.92 |

Table 7: MSE ($\times 10^{-2}$) on Mocap Walk *w.r.t.* the number of clusters $K$.

| 31 nodes | 1 | 2 | 3 | 4 | 5 | 6 | 7 | 8 |
|----------|------|------|------|---------|-----|------|------|------|
| MSE | 19.8 | 16.8 | 10.1 | **8.1** | 8.5 | 10.5 | 11.2 | 14.9 |

Table 8: MSE ($\times 10^{-2}$) on Protein MD *w.r.t.* the number of clusters $K$.

| 855 nodes | 1 | 5 | 10 | 15 | 20 | 25 |
|-----------|-------|-------|-------|-----------|-------|-------|
| MSE | 2.132 | 2.234 | 2.127 | **1.843** | 2.189 | 2.245 |

We have these investigations: **1.** On all datasets, the performance degenerates when $K = 1$, since all nodes in the system are pooled into one cluster and therefore there are no learnable cluster assignments. It verifies the necessity of modeling hierarchies in multi-body systems. **2.** The systems with larger scale enjoys larger $K$ in practice. It indicates that for the systems with larger number

of nodes, it is beneficial to choose larger $K$ to better model their complex hierarchies. **3.** For the Complex (3,3) system, it is interesting that the best performance is obtained when $K = 3$, since it contains 3 disjoint complexes. This implies that it is also possible to choose $K$ by some prior knowledge assessed from data.

## D.2 The choice of internal and external modules

In this subsection we provide ablation study that compares the performance of different choices between internal/external EMMP/EGNN. The experimental results are exhibited in Table 9.

Table 9: MSE ($\times 10^{-2}$) on two motion capture datasets and two $M$-Complex systems.

| Internal | External | Mocap Walk | Mocap Run | Complex (3, 3) | Complex (5, 5) |
|----------|----------|------------|-----------|----------------|----------------|
| EGNN | EGNN | 22.3 | 42.5 | 12.51 | 15.77 |
| EMMP | EGNN | 8.5 | 21.9 | **11.58** | 14.42 |
| EMMP | EMMP | **8.1** | **21.1** | 11.82 | **14.36** |

We have the following observations:

- When applying EMMP in either internal or external message passing, the performance consistently improves against EGNN. This verifies that the proposed EMMP is potentially more advantageous on modeling interactions, which aligns with our theoretical analysis that EMMP is more expressive than EGNN (c.f. Theorem 1).

- Compared with external EMMP, more significant improvements are obtained when applying EMMP as the internal message passing layers (e.g., $22.3 \rightarrow 8.5$ on MocapWalk). Note that the internal message passing layers are those right before our pooling layer, which are responsible for passing and aggregating messages towards the high-level cluster nodes. Therefore, we speculate the reason might be that compared with the flat message passing layers (the external EMMPs), the internal EMMPs require much higher expressivity and capacity since they need to fuse the message of all nodes towards their corresponding cluster nodes.

- In the Complex (3, 3) scenario, changing from EGNN to EMMP in external message passing slightly affects the performance, probably because the interactions between nodes in $M$-complex are Coulomb forces which can be well covered by EGNN. Nevertheless, on the mocap dataset where interactions are much more complicated, leveraging EMMP is consistently more advantageous over EGNN.

## D.3 The hierarchy ablation study with identity assignments.

We summarize in Table 10 the results of more ablation studies on all datasets (simulation, mocap, and protein), where EGHN w/o hier is implemented by setting the cluster assignment to identity, *i.e.*, $\mathbf{s}_i = \mathbf{1}_i$.

Table 10: MSE ($\times 10^{-2}$) on five datasets with and without identity assignments.

| | Complex (3,3) | Complex (5,5) | Mocap Walk | Mocap Run | Protein MD |
|---|---------------|---------------|------------|-----------|------------|
| EGHN | **11.58** | **14.42** | **8.5** | **25.9** | **1.8**4 |
| EGHN w/o hier | 12.24 | 15.18 | 21.9 | 42.1 | 2.00 |

As illustrated in Table 10, the hierarchical structure is consistently beneficial to the model performance across $M$-complex simulation, Motion Capture, and Protein MD. This supports the validity and efficacy of our designed equivariant hierarchy module.

## E   Training time comparison

We evaluate the training time on simulation and motion capture datasets for the baselines and EGHN. Table 11 depicts the average training time per epoch (in seconds). All models are trained on a NVIDIA V100 GPU.

Table 11: The average training time per epoch (in seconds) on two datasets.

|  | MPNN [3] | TFN [7] | SE(3)-Tr. [2] | EGNN [6] | GMN [4] | EGHN |
|---|---|---|---|---|---|---|
| Complex (3, 3) | 1.21 | 7.81 | 23.25 | 1.45 | 1.58 | 1.69 |
| MocapWalk | 0.92 | 6.85 | 18.96 | 1.21 | 1.49 | 1.41 |

EGHN is almost as efficient as EGNN and GMN, while only adding marginal computational overhead compared to MPNN, since the computations related to equivariance and pooling are efficient. The irreps-based methods TFN and SE(3)-Transformer yield significantly longer training time.

## F   Comparison with additional baselines

We also compare with SEGNN [1] on $M$-complex systems. The results are in Table 12. SEGNN performs better than EGNN particularly when the system is large (*e.g.*, on (5, 10) or (10, 10)). Still, EGHN consistently outperforms these baselines by a significant margin.

Table 12: Prediction error ($\times 10^{-2}$) on various types of simulated datasets. The "Multiple System" contains $J = 5$ different systems. For each column, $(M, N/M)$ indicates that each system contains $M$ complexes of average size $N/M$. Results averaged across 3 runs. "OOM" denotes out of memory.

|  | Single System | | | | Multiple Systems | | | |
|---|---|---|---|---|---|---|---|---|
|  | (3, 3) | (5, 5) | (5, 10) | (10, 10) | (3, 3) | (5, 5) | (5, 10) | (10, 10) |
| EGNN [6] | 12.69 | 15.37 | 15.12 | 14.64 | 13.33 | 15.48 | 15.29 | 15.02 |
| SEGNN [1] | 14.04 | 15.62 | 15.01 | 14.31 | 13.88 | 16.01 | 15.41 | 14.78 |
| EGHN | **11.58** | **14.42** | **14.29** | **13.09** | **12.80** | **14.85** | **14.50** | **13.11** |

## G   More Visualizations

In this section, we provide more visualization results. Figure 10, Figure 11, and Figure 12 illustrate more visualization examples on (5, 5, 1) of the simulation dataset, walking on the motion capture dataset, and the MD dataset, respectively.

We further provide more predictions and pooling results of EGHN in Fig. 9. It is observed that EGHN gives accurate predictions with desirable pooling assignments.

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

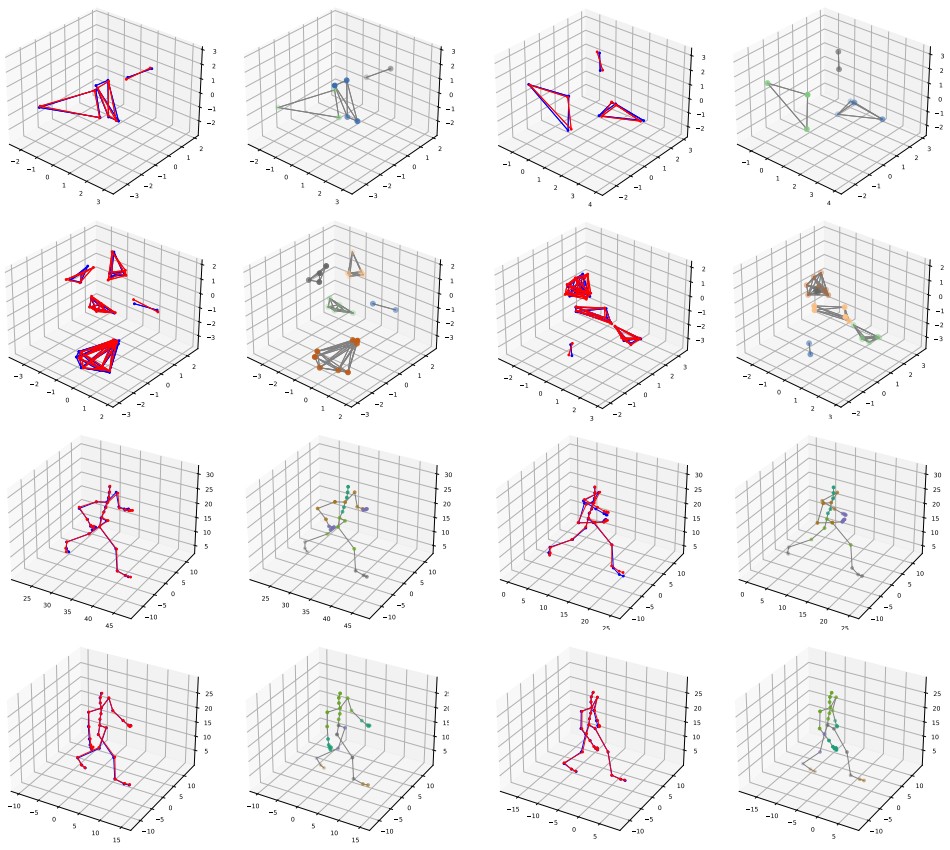

Figure 9: More visualizations and pooling results. Ground truth in red. The prediction of EGHN in blue.

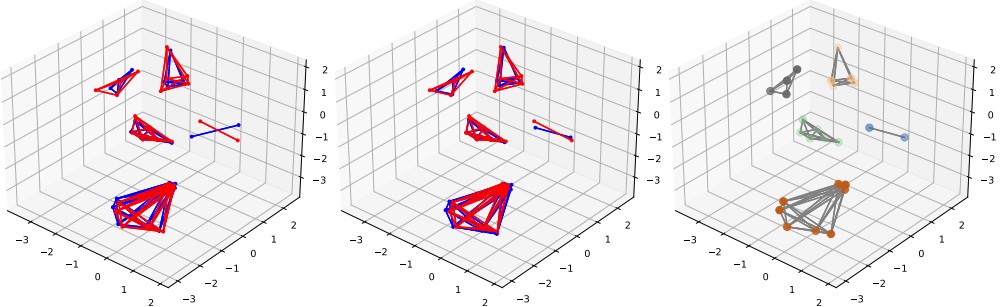

Figure 10: Visualization on $M$-complex dataset. *Left*: the prediction of EGNN. *Middle*: the prediction of EGHN. *Right*: the pooling results of EGHN with each color indicating a cluster. Ground truth in red, and prediction in blue. Best viewed by colour printing and zooming in.

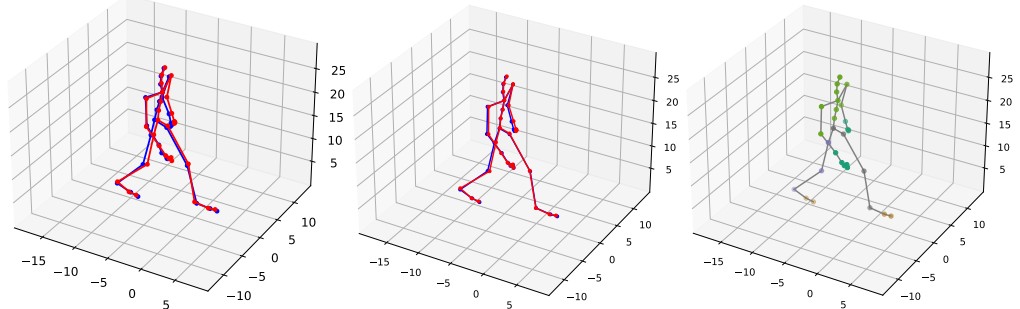

Figure 11: Visualization on Mocap Walk. *Left*: the prediction of EGNN. *Middle*: the prediction of EGHN. *Right*: the pooling results of EGHN with each color indicating a cluster. Ground truth in red, and prediction in blue. Best viewed by colour printing and zooming in.

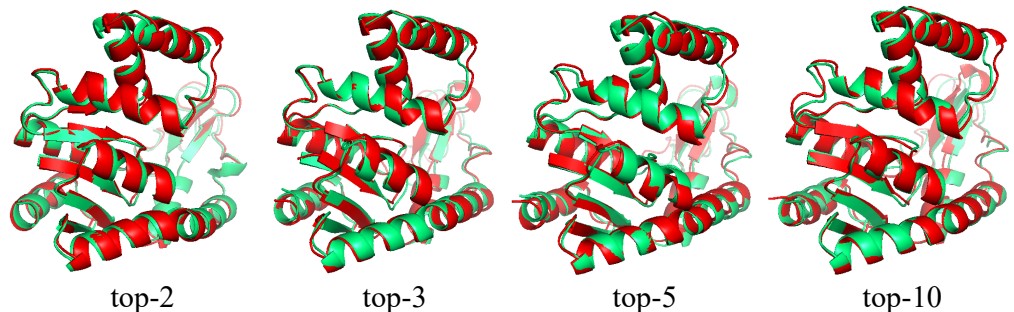

| top-2 | top-3 | top-5 | top-10 |

Figure 12: More visualizations on protein MD. Ground truth in red. The prediction of EGHN in green.

[5] Jonas Köhler, Leon Klein, and Frank Noé. Equivariant flows: sampling configurations for multi-body systems with symmetric energies. *arXiv preprint arXiv:1910.00753*, 2019. 3

[6] Victor Garcia Satorras, Emiel Hoogeboom, and Max Welling. E(n) equivariant graph neural networks. *arXiv preprint arXiv:2102.09844*, 2021. 1, 3, 7

[7] Nathaniel Thomas, Tess Smidt, Steven Kearnes, Lusann Yang, Li Li, Kai Kohlhoff, and Patrick Riley. Tensor field networks: Rotation-and translation-equivariant neural networks for 3d point clouds. *arXiv preprint arXiv:1802.08219*, 2018. 3, 7